# Development of a new cell isolation device FlowMagic™

**Tomoyuki Yoshida**[1]*, **Yoshiaki Sakamoto**[1], **Aya Tsuruta**[1], **Risa Kimura**[1], **Nozomi Shiozawa**[2], **Takeyuki Kotaka**[1]*

**1** H.U. Group Research Institute G.K., Akiruno, Tokyo, Japan, **2** H.U. Cells, Inc., Akiruno, Tokyo, Japan

☯ These authors contributed equally to this work.
* takeyuki.kotaka@hugp.com (TK); tomoyuki.yoshida@hugp.com (TY)

## Abstract

Isolation of human peripheral blood mononuclear cells (PBMCs) from blood typically involves a density gradient medium during density centrifugation. The problem of increasing red blood cell (RBC) and granulocyte (GRA) contamination during PBMC isolation as the elapsed time after blood collection increases remains unresolved. As a countermeasure against RBC contamination, hemolysis treatment is available; however, these extra steps are laborious, time-consuming, and could introduce artifacts. To overcome this challenge, we developed a novel isolation device, FlowMagic™, which features a proprietary two-layer insert structure designed to prevent RBC and GRA contamination during PBMC isolation from blood. The efficacy of this method was evaluated by isolating PBMCs from donors and analyzing immune cell populations by flow cytometry. Compared to SepMate (median (Q50) = 11.0, interquartile ranges (IQR): 8.8–19.5; p < 0.01) and Lymphoprep methods (Q50 = 9.3, IQR: 6.6–13.5; p < 0.01), FlowMagic™ achieved significantly greater reduction in RBC contamination to below detectable limits (Q50 = 0.0, IQR: 0.0–0.0), with sustained efficacy observed up to 72 hours post-collection. Additionally, the FlowMagic™ method (Q50 = 2.5, IQR: 0.5–3.4, at 48 hours, median = 4.5, IQR: 2.1–10.3, at 72 hours, respectively) significantly reduced GRA contamination compared with the SepMate (Q50 = 12.0, IQR: 7.8–25.5, at 48 hours, Q50 = 27.5, IQR: 12.3–29.0, at 72 hours, respectively; p < 0.01) and Lymphoprep methods (Q50 = 10.5, IQR: 6.9–19.8, at 48 hours, Q50 = 17.5, IQR: 13.3–23.5, at 72 hours, respectively; p < 0.01) at 48 and 72 hours after blood collection. Furthermore, the recovery rates of CD3 +, CD4 +, CD8 +, CD19 +, and CD16/56 + cells in the FlowMagic™-isolated PBMCs (Q50 = 8.6, 5.9, 2.5, 1.3, and 1.9, respectively) were significantly improved compared to those in SepMate- (Q50 = 2.2, 1.5, 0.7, 0.4, and 0.5, respectively; p < 0.01) and Lymphoprep-isolated PBMCs (Q50 = 2.4, 1.5, 0.8, 0.6, and 0.8, respectively; p < 0.01), even at 48 hours after blood collection. These findings suggest that the PBMC isolation method using FlowMagic™ is advantageous in preventing RBC and GRA contamination for research, diagnostic, and clinical applications.

**Data availability statement:** All relevant data are within the manuscript and its Supporting information files.

**Funding:** The author(s) received no specific funding for this work.

**Competing interests:** Dr. Tomoyuki Yoshida, Dr. Takeyuki Kotaka, Mr. Yoshiaki Sakamoto, Ms. Aya Tsuruta, and Ms. Risa Kimura are employees of H.U. Group Research Institute G.K.. Ms. Nozomi Shiozawa is an employee of H.U. Cells, Inc. The funder provided support in the form of salaries for authors [TY, TK, YS, AT, RK, NS], but did not have any additional role in the study design, data collection and analysis, decision to publish, or preparation of the manuscript. The specific roles of these authors are articulated in the 'Author Contributions' section. The authors have filed a PCT (Patent Cooperation Treaty) application related to the methods and findings described in this study. The patent application number is PCT/JP2024/035622 and is currently pending. This does not alter the authors' adherence to PLOS ONE policies on sharing data and materials, and all data necessary to replicate the findings are available as described in the manuscript.

## Introduction

The isolation of human peripheral blood mononuclear cells (PBMCs) from whole blood in laboratory tests, clinical trials, and basic scientific research is a crucial preprocessing step for various assays, such as enzyme-linked immunosorbent spot (ELISPOT) assays [1,2], proliferation assays [3], flow cytometry [4], and cytometry by time-of-flight (CyTOF) [5]. In these assays, isolated cells should exhibit high viability and be substantially free from RBC, granulocyte and platelet contamination.

PBMCs are primarily isolated by density gradient centrifugation [6]. While the standard method for PBMC isolation is Ficoll-Paque gradient centrifugation, other PBMC isolation devices have recently become commercially available, due to their faster and simpler processes. These devices have a shorter centrifugation time with a brake, which is shorter than the Ficoll-Paque technique. The BD Vacutainer cell preparation tube (CPT) is an evacuated tube containing an anticoagulant and a cell separation medium containing a polyester gel and a density gradient liquid [7,8]. The Greiner Bio-One LeucoSep tube has a porous membrane frit that isolates the density gradient from the whole blood sample [8,9]. In addition, as a second-generation cell isolation device, the STEMCELL Technology SepMate tubes contain an insert that creates a barrier between the density gradient medium and blood, thus eliminating the need for careful blood layering and allowing mononuclear cells to be easily harvested with a simple pour [7,8].

Though there are many technologies and techniques that increase the ease of PBMC isolation, these methods are plagued by low level of contaminating RBCs [10]. Moreover, when PBMCs are isolated on a large scale, as with most ex vivo adoptive immunotherapy methods, the amount of contaminating RBC increases even further. The presence of RBC contamination in a PBMC sample may lead to inaccurate PBMC concentration measurements, which can be detrimental to downstream experiments performed with these cells. To resolve this issue, an RBC lysis protocol can be used to reduce RBC contamination, however, these extra steps are laborious, time-consuming, and could be a source of artifacts.

Granulocyte (GRA) is also a major cause of contamination during PBMC processing. GRA usually pellet with RBCs. However, prolonged storage of whole blood at room temperature for 24–48 hours prior to processing, due to transportation or facility limitations, has been shown to increase GRA contamination [11]. Hence, delaying PBMC processing influences the sample quality, which affects downstream assays. Prolonged storage-induced GRA contamination has been shown to correlate with reduced T cell function, decreased interferon-γ ELISPOT spot counts, and altered T cell metabolic pathway activities [11–13]. Thus, it is important to avoid RBC and GRA contamination during PBMC processing.

The purpose of this study is to report on the performance of a novel cell isolation device, FlowMagic™, which uses proprietary technology to prevent RBC and GRA contamination during PBMC isolation from blood. We evaluated the PBMC recovery rates and purity using an automated cell counter and flow cytometry.

## Materials and methods

### PBMC isolation

Venous blood was collected from healthy volunteers into sodium-heparin tubes (Termo Corporation) after obtaining written consent in accordance with guidelines for biomedical research involving human subjects. The study protocol was approved by the Institutional Review Board of the Ethical Review Committee of H.U. Group Holdings, Inc. (Approval number: 24-015-00, 24-015-02). The research period extended from 15/04/2024, to 31/03/2026. Written informed consent was obtained from volunteer participants prior to each blood collection procedure. Recruitment of volunteer participants for this study commenced on 25/04/2024 and concluded on 15/11/2024.

Fresh whole blood samples were incubated at room temperature (20–24°C) immediately after collection. Parallel PBMC isolations were performed by three different techniques within 24, 48, and 72 hours after blood collection from the same volunteers. Isolation with Lymphoprep (density of 1.077 g/mL; STEMCELL Technology) with FlowMagic™ tubes (in-house development), Lymphoprep with SepMate tubes (STEMCELL Technology), and Lymphoprep (STEMCELL Technology) alone in 50 mL polypropylene tubes (Wako) were performed according to in-house protocols and the manufacturer's instructions.

FlowMagic™ 3D CAD images, FlowMagic™ isolation workflow and the three isolation protocols are summarized in Figs 1 and 2. For the FlowMagic™ technique, 15 mL of Lymphoprep was added to the FlowMagic™ tube, and centrifuged (1,200 g, 10 min, brake on) for 5 minutes. Heparinized blood was diluted twice with phosphate-buffered saline (PBS; FUJIFILM Wako Pure Chemical Corporation) containing 2% fetal bovine serum (FBS) and injected into the nozzle hole on upper insert of FlowMagic™. After centrifugation (1,200 g, 10 min, brake on), and pressing down the floating upper insert with a plastic tip, PBMCs are collected by pouring the supernatant into a 50 mL polypropylene tube (Wako). For the Sep-Mate technique, heparinized blood was diluted twofold with PBS containing 2% FBS and layered on top of Lymphoprep. After centrifugation (1,200 g, 10 min, brake on), PBMCs were collected by pouring the supernatant into a 50 mL polypropylene tube. For the Lymphoprep technique, heparinized blood is diluted twofold with PBS containing 2% FBS and layered on top of Lymphoprep in the 50 mL polypropylene tube. After centrifugation (1,200 g, 20 min, no brake), PBMCs were collected using a plastic pipette and stored in a 50 mL polypropylene tube. Collected PBMCs were washed twice with PBS (FUJIFILM Wako Pure Chemical Corporation) containing 2% FBS (500 g, 10 min and 5 min, brake on). Cell counts were performed using a Microsemi LC-710 (HORIBA) automated cell counter.

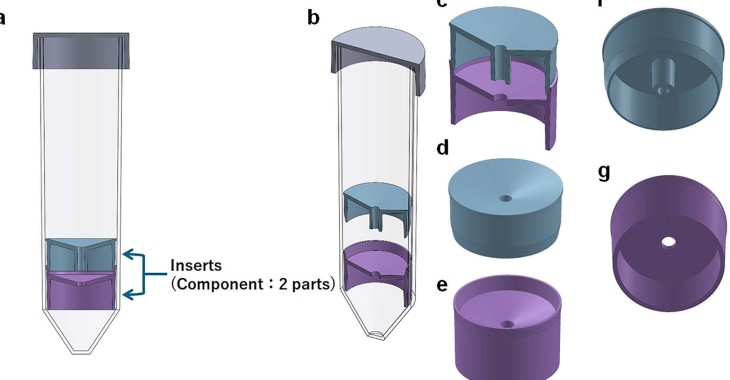

**Fig 1. 3D CAD images of FlowMagic™.** (a) Cross-sectional view of FlowMagic™. The components of inserts in FlowMagic™ are divided into two parts. (b) The oblique cross-sectional view of the floating upper insert and the fixed lower insert in FlowMagic™. (c) The oblique cross-sectional view of the inserts. (d) Top view of the upper insert. (e) Top view of the lower insert. (f) Bottom view of the upper insert. (g) Bottom view of the lower insert.

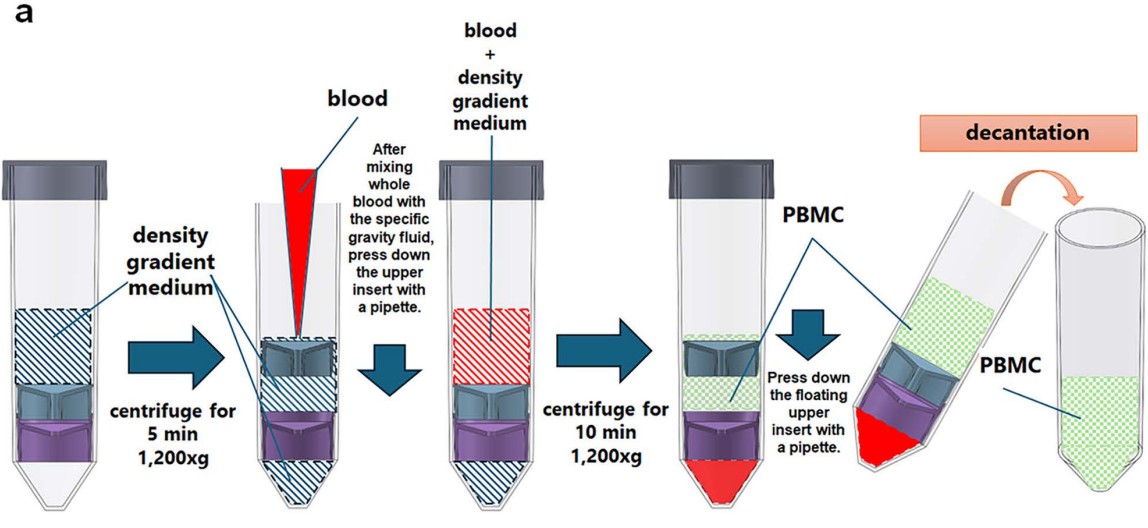

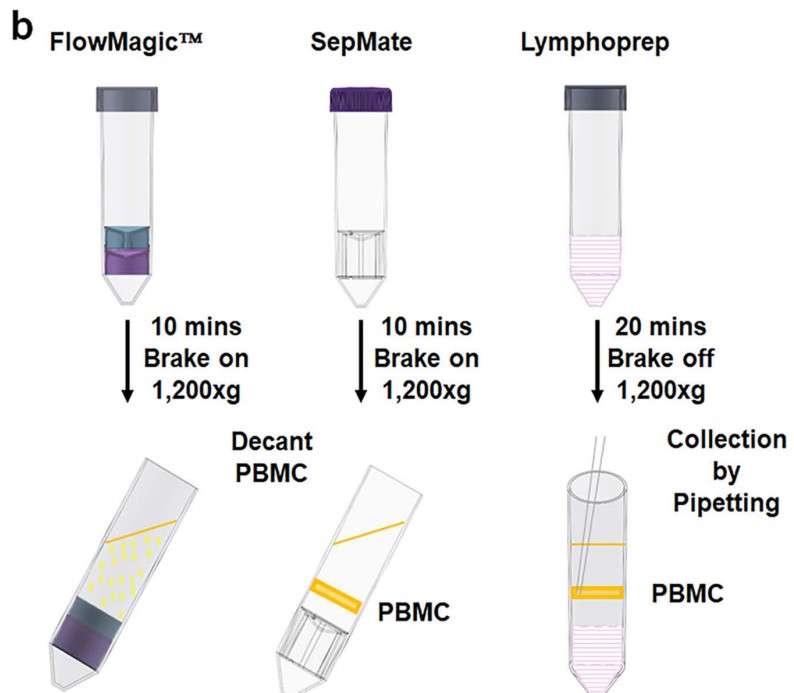

**Fig 2. FlowMagic™ isolation workflow and schematic presentation of the three isolation techniques.** (a) 15 mL of Lymphoprep was added to the FlowMagic™ tube. Whole blood is diluted twofold with PBS, mixed with a density gradient medium, and compressed with a pipette using the floating upper insert. The mixture is then centrifuged for 10 minutes. PBMCs are collected by decantation in the tube. (b) Comparison of between FlowMagic™, SepMate and Lymphprep PBMC isolation methods. Centrifugation time and speed was indicated prior to collection of PBMCs. PBMCs are decanted for FlowMagic™ and SepMate while PBMC was collected for Lymphoprep by pipetting. The orange area for SepMate and Lymphprep indicates the PBMC layer.

## Immunophenotyping of PBMC populations

Immunophenotyping was performed by flow cytometry. PBMC samples were stained using Multitest 6-Color TBNK staining kit (Becton Dickinson). Samples were measured with a BD FACSLyric system (BD Bioscieneces) and analyzed with FlowJo software v10.

## Statistical analysis

To assess the differences between the FlowMagic™ method and two commercially available methods, pairwise comparisons were conducted: FlowMagic™ versus SepMate and FlowMagic™ versus Lymphoprep. Wilcoxon signed-rank test with Bonferroni correction was used for multiple comparisons as a non-parametric alternative. *p*-values less than of equal to 0.05 were considered statistically significant for all statistical tests. All statistical analyses were performed using Python version 3.13.5.

## Results

### FlowMagic™ reduces RBC and GRA contamination during PBMC isolation

Three isolation methods were performed: FlowMagic™ tube with Lymphoprep, denoted as FlowMagic™; SepMate tube with Lymphoprep, denoted as SepMate; and normal tube with Lymphoprep, denoted as Lymphoprep. Fig 1 shows the patent-pending 3D CAD images of FlowMagic™, which consists of two inserts in the 50 mL polypropylene tube. It differs from previous cell isolation devices in that the upper insert is designed to float in the tube during centrifugation. Fig 2 shows FlowMagic™ isolation workflow and a comprehensive schematic comparison of the three protocols regarding centrifugation time and speed, and collection protocol. First, we examined the recovery of PBMCs from whole blood from ten volunteers by using these three different isolation methods. The state of fractionation from whole blood collected at 24, 48, and 72 hours after blood collection using each method is shown in Fig 3. The upper Layer contains plasma, isolation medium, and PBMCs. The middle Layer (interface) contains a few RBCs and some GRAs in FlowMagic™. The lower Layer predominantly contains RBCs and GRAs in FlowMagic™.

Fig 4a–c shows changes in white blood cell (WBC), RBC, and GRA counts, respectively, from whole blood samples processed at 24, 48, and 72 hours using three isolation methods: FlowMagic™, SepMate, and Lymphoprep. At 24 hours after blood collection, no RBC contamination was observed in all three methods (Figs 3a and 4b).

FlowMagic™ achieved significantly greater reduction in RBC contamination to below detectable limits even up to 72 hours after collection (Q50 = 0.0, IQR: 0.0–0.0) compared with SepMate (Q50 = 11.0, IQR: 8.8–19.5; p < 0.01) and Lymphoprep methods (Q50 = 9.3, IQR: 6.6–13.5; p < 0.01) (Fig 4b). Additionally, the FlowMagic™ method (Q50 = 2.5, IQR: 0.5–3.4 at 48 hours, Q50 = 4.5, IQR: 2.1–10.3 at 72 hours) significantly reduced GRA contamination compared with the SepMate (Q50 = 12.0, IQR: 7.8–25.5 at 48 hours, Q50 = 27.5, IQR: 12.3–29.0 at 72 hours; p < 0.01, p < 0.01, respectively) and Lymphoprep methods (Q50 = 10.5, IQR: 6.9–19.8 at 48 hours, Q50 = 17.5, IQR: 13.3–23.5 at 72 hours; p < 0.01, p < 0.01, respectively) at 48 and 72 hours after blood collection. (Fig 4c). A summary of the main PBMC composition data is provided in S1 Table. These results suggest the efficacy of FlowMagic™ to reduce RBC and GRA contamination.

Next, we evaluated the purity and composition of PBMCs isolated from whole blood the three different isolations methods (Fig 5). PBMCs isolated with FlowMagic™ (Q50 = 67.2, IQR: 61.5–69.6, p < 0.01) exhibited significantly higher lymphocyte purity rates compared to those isolated using SepMate (Q50 = 56.6, IQR: 50.0–62.2, p < 0.01) and Lymphoprep (Q50 = 56.0, IQR: 49.0–59.4, p < 0.01) at 72 hours after blood collection (Fig 5a). PBMCs isolated with FlowMagic™ also yielded significantly more monocytes than SepMate and Lymphoprep at 72 hours after blood collection (Fig 5b). The contamination rate of GRA in the isolated PBMC was significantly reduced when FlowMagic™ isolation method was used at 48 hours and 72 hours after blood collection (Fig 5c). A summary of the lymphocyte composition data is provided in S2 Table.

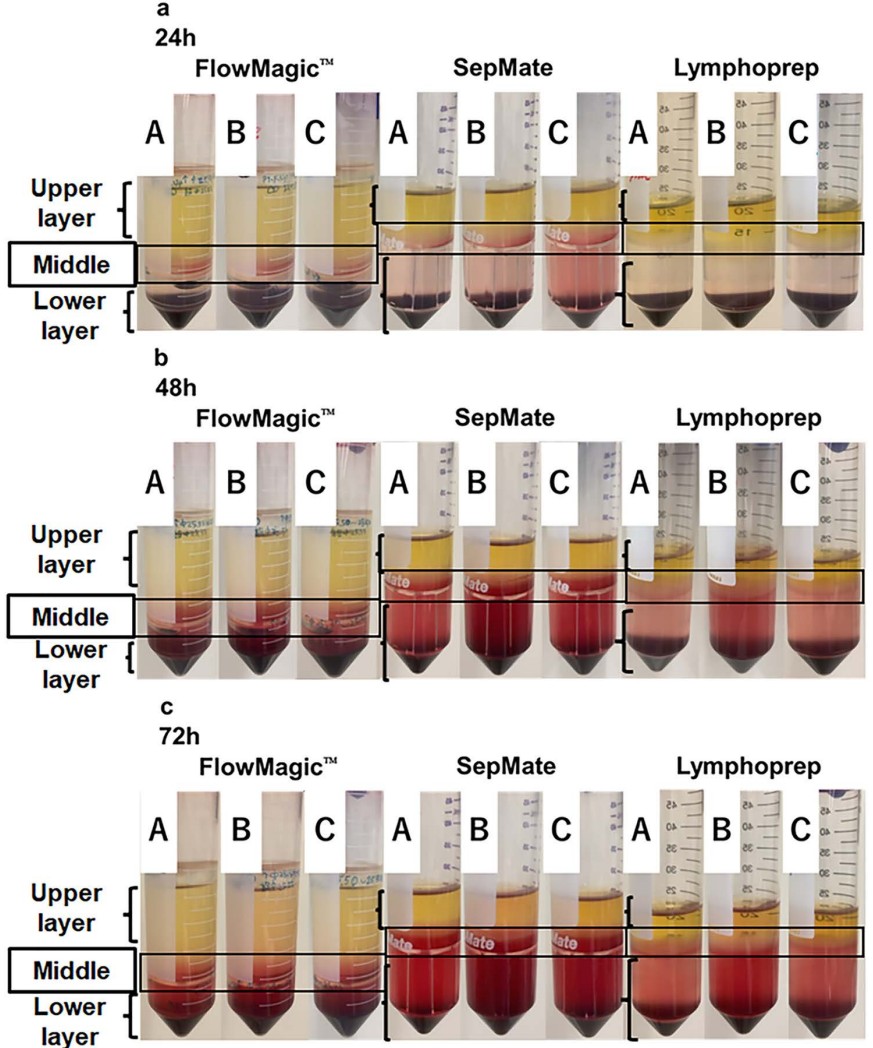

**Fig 3. Time-dependent changes in the state of isolation of blood.** Whole blood was isolated using Lymphoprep in a FlowMagic™ tube, SepMate tube, and a normal tube. Sample A, B, and C refer to three different donors. Different layers were labelled. The upper Layer contains plasma, isolation medium, and PBMCs in FlowMagic™. The middle Layer (Interface) contains a few RBCs and some GRAs in FlowMagic™. The lower Layer predominantly contains RBCs and GRAs in FlowMagic™. (a) 24 hours after blood collection. (b) 48 hours after blood collection. (c) 72 hours after blood collection.

## Population composition

We then assessed the composition of leukocytes that are of interest to functional assays in the isolated PBMC fractions, namely: T-cells (CD3+), helper T-cells (CD4+), cytotoxic T-cells (CD8+), B-cells (CD19+), and NK cells (CD16/56+) by flow cytometry. The composition of the isolated PBMC populations varied between volunteers, but in general the percentage of CD3+, CD4+, CD8+ CD19+, and CD16/56+ cells isolated from FlowMagic™ were greater than from SepMate and Lymphoprep (Fig 6a-6e) up to 72 hours after blood collection. This improvement in enrichment of leukocytes from FlowMagic™ when compared to SepMate and Lymphoprep was more significant at 48 hours and 72 hours after blood collection (Fig 6a-6e). There is a notable drop though in the percentage of leukocytes at 48 hours and 72 hours after blood collection, which may be due to the accumulation of a small amount of GRA contamination (Fig 5c). A summary of the lymphocyte

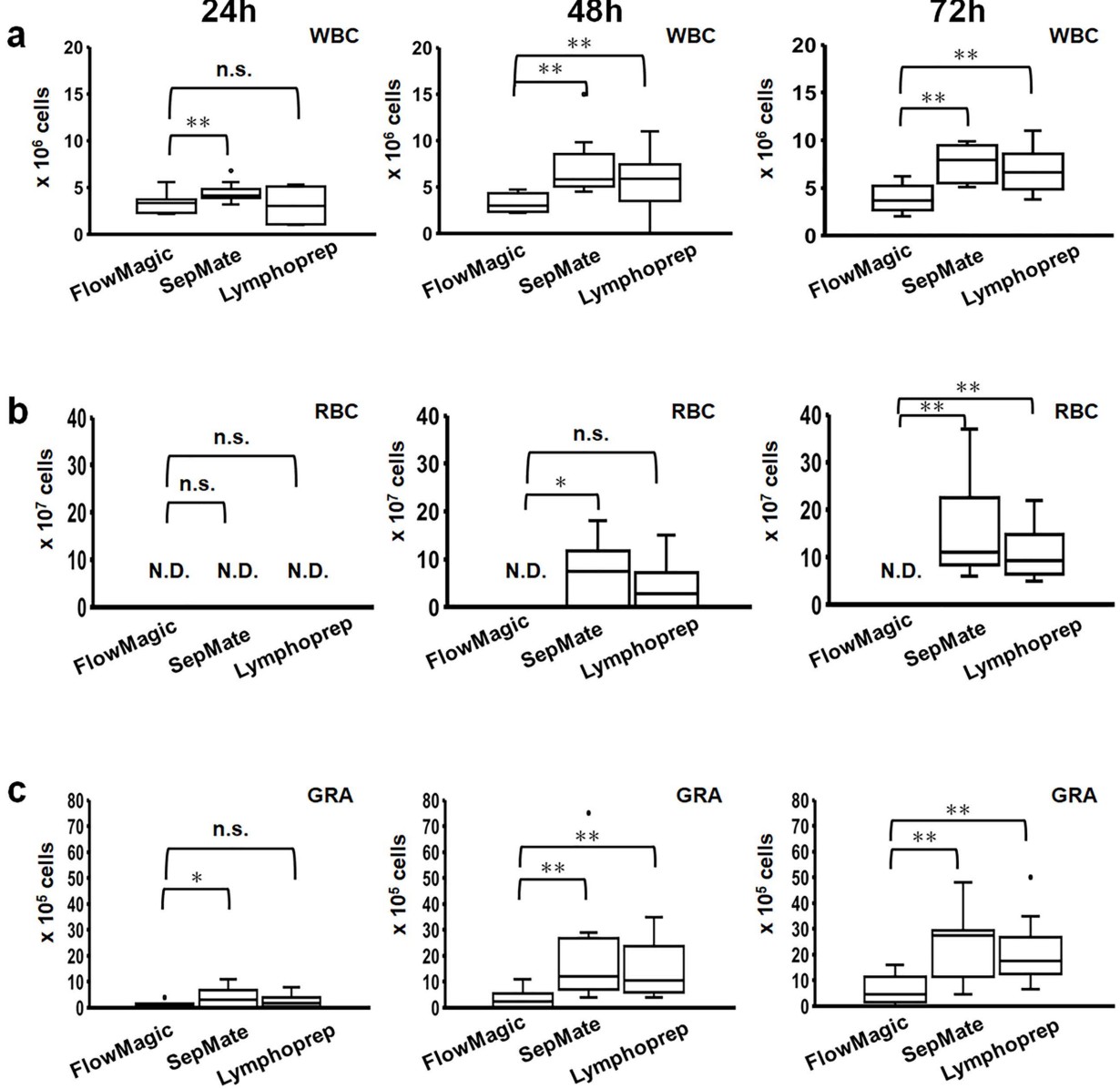

**Fig 4. Comparative performance of cell isolation methods (FlowMagic™, SepMate, Lymphoprep) following extended whole blood storage.** Cell counts of (a) WBC, (b) RBC, and (c) GRA are shown at 24, 48, and 72 hours after collection. Box plots show median (center line), interquartile ranges (boxes, 25th to 75th percentiles), whiskers extending to 1.5×IQR, and outliers as individual dots. Statistical significance was assessed using Wilcoxon signed-rank test (*p<0.05, **p<0.01, and n.s., not significant). N=10. N.D.: Not detected; WBC: White blood cell; RBC: Red blood cell; GRA: Granulocyte; IQR: interquartile ranges.

and leukocyte composition data is provided in S2–S3 Tables respectively. Overall, these results suggest the efficacy of the FlowMagic™ isolation method, for enrichment of leukocytes from blood.

## Discussion

In this paper, we reported on the efficacy of the new cell isolation device, FlowMagic™. Three PBMC isolation techniques were evaluated, focusing on cell recovery and population composition. The techniques comprised the isolation by the

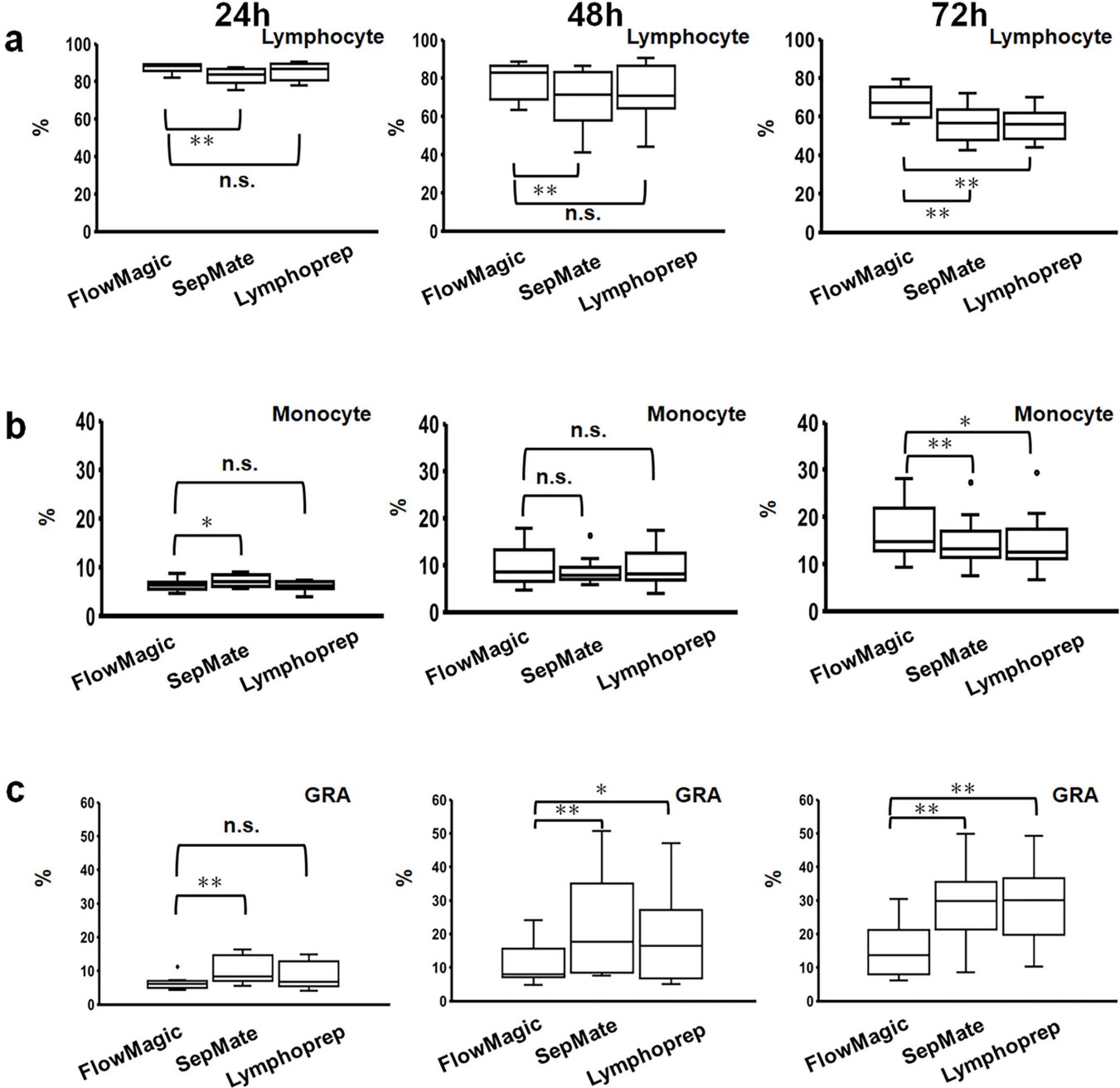

**Fig 5. Comparative recovery rates of cell isolation methods (FlowMagic™, SepMate, Lymphoprep) following extended whole blood storage.** Cell counts of (a) Lymphocyte, (b) Monocyte, and (c) GRA are shown at 24, 48, and 72 hours after collection. Box plots show median (center line), inter-quartile ranges (boxes, 25th to 75th percentiles), whiskers extending to 1.5×IQR, and outliers as individual dots. Statistical significance was assessed using Wilcoxon signed-rank test (*p<0.05, **p<0.01, and n.s., not significant). N=10. GRA: Granulocyte; IQR: interquartile ranges.

FlowMagic™ device with Lymphoprep, isolation by SepMate tubes with Lymphoprep, and only Lymphoprep. Lymphoprep isolation is the most attractive approach from a cost perspective, but in practice, this technique is relatively laborious and takes longer than the other two methods, and is plagued by RBC and GRA contamination. SepMate increases the yield

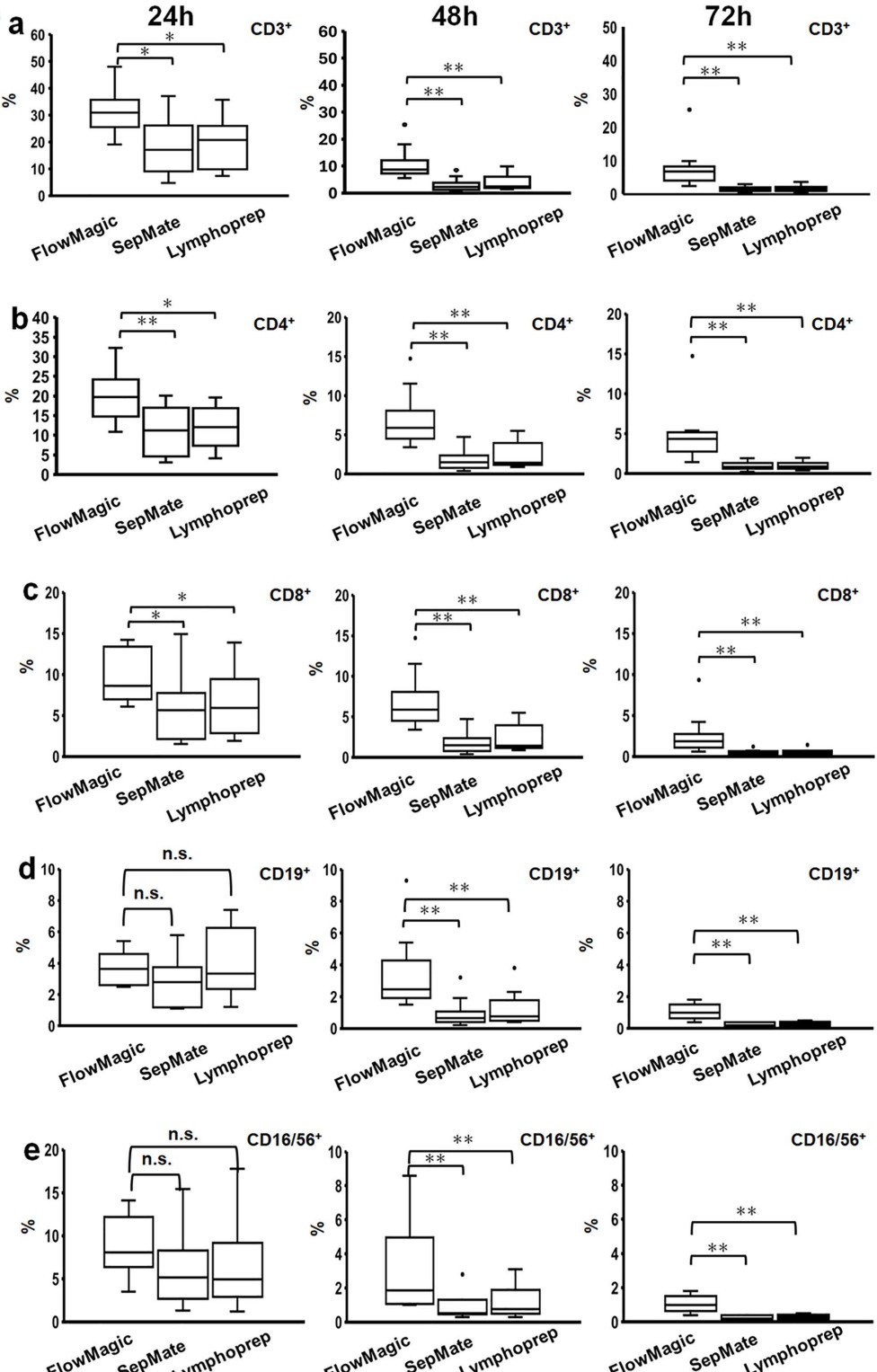

**Fig 6. PBMC composition by flow cytometry of cell isolation methods (FlowMagic™, SepMate, Lymphoprep) following extended whole blood storage.** Cell counts of (a) CD3+, (b) CD4+, (c) CD8+, (d) CD19+, and (e) CD16/56+ cells are shown at 24, 48, and 72 hours after collection. PBMCs were isolated and stained with Multitest 6-Color TBNK consisting of anti-CD45 (leukocytes), -CD3 (T-cells), -CD4 (helper T-cells), -CD8 (cytotoxic T-cells),

-CD20 (B-cells), and -CD16/56 (NK-cells). Box plots show median (center line), interquartile ranges (boxes, 25th to 75th percentiles), whiskers extending to 1.5×IQR, and outliers as individual dots. Statistical significance was assessed using the Wilcoxon signed-rank test (*p < 0.05, **p < 0.01, and n.s., not significant). N = 10. PBMC: peripheral blood mononuclear cell; IQR: interquartile ranges.

of WBC and reduces processing time, but has problems with RBC and GRA contamination. This contamination becomes more pronounced during extended delays in blood processing. Thus, we propose that FlowMagic™ is a superior alternative method because it yields significantly lower RBC and GRA contamination than SepMate and Lymphoprep isolation methods even on isolated PBMCs processed at 48 and 72 hours after blood collection. Moreover, the recovery rates of CD3$^+$, CD4$^+$, CD8$^+$, CD19$^+$, and CD16/56$^+$ cells in FlowMagic™-isolated PBMCs were significantly improved compared to SepMate- and Lymphoprep-isolated PBMCs even at 48 and 72 hours after blood collection. Based on these findings, FlowMagic™ has the potential to address RBC and GRA contamination that occurs over lengthy delays of sample processing after blood collection, particularly at 48–72 hours post-blood collection.

In overseas clinical trials, the transportation time after blood collection can take 24–48 hours, and long transportation time can degrade the quality of PBMCs [11,12]. Even in such cases, FlowMagic™ can be an effective solution for isolating PBMC with high purity. Overall, the FlowMagic™ method demonstrated exceptional efficacy, achieving complete removal of RBC contamination and a substantial reduction of GRA contamination, surpassing the performance of commonly used isolation techniques. Further studies are needed to evaluate how PBMCs isolated using the FlowMagic™ device affect cell function assays that are commonly plagued by RBC contamination.

During PBMC isolation using SepMate method and Lymphoprep method, whole blood is layered on top of the density gradient medium, hence the whole blood and the density gradient medium are not mixed [7,8]. In contrast, the FlowMagic™ isolation method mixes the density gradient medium and whole blood due to the lowering of the upper insert before centrifugation (Fig 2a). Due to this step in the FlowMagic™ isolation protocol, one disadvantage of FlowMagic™ is that it cannot simultaneously separate plasma during PBMC isolation. The inability of FlowMagic™ to isolate plasma simultaneously may be relevant for translational settings. Functional downstream assays (e.g., cytokine release, ELISPOT) are not yet presented and should be acknowledged as needed future work.

## Conclusion

We propose that FlowMagic™ is a useful tool for isolating PBMCs with low levels of RBC and GRA contamination from samples with processing delays of 24–72 hours after blood collection compared to Lymphoprep alone or other commercial cell isolation devices such as SepMate. Our results have reported the efficacy of PBMC isolation from whole blood, however further extensive studies using *in vitro* cell functional assays are clearly needed. The present study suggests that the newly developed cell isolation device, FlowMagic™, may be useful in fields such as laboratory testing and cell culture systems.

## Supporting information

**S1 Table. Summary of PBMC composition isolated by three different methods from whole blood after 24-hour storage.** Data from 10 volunteers. Values are shown as means ± standard deviation. The Wilcoxon signed-rank test was used with Bonferroni correction for multiple comparisons as a non-parametric alternative. A significance level of 0.05 was used for all statistical tests. PBMC: peripheral blood mononuclear cell; WBC: white blood cell; RBC: red blood cell; GRA: granulocytes.
(TIF)

**S2 Table. Summary of PBMC composition isolated by three different methods from whole blood after 48-hour storage.** Data from 10 volunteers. Values are shown as means ± standard deviation. The Wilcoxon signed-rank test was

used with Bonferroni correction for multiple comparisons as a non-parametric alternative. A significance level of 0.05 was used for all statistical tests. PBMC: peripheral blood mononuclear cell; WBC: white blood cell; RBC: red blood cell; GRA: granulocytes.

(TIF)

**S3 Table. Summary of PBMC composition isolated by three different methods from whole blood after 72-Hour storage.** Data from 10 volunteers. Values are shown as means ± standard deviation. The Wilcoxon signed-rank test was used with Bonferroni correction for multiple comparisons as a non-parametric alternative. A significance level of 0.05 was used for all statistical tests. PBMC: peripheral blood mononuclear cell; WBC: white blood cell; RBC: red blood cell; GRA: granulocytes.

(TIF)

## Acknowledgments

The authors would like to give special thanks to the members of the Institutional Review Board of the Ethical Review Committee of H.U. Group Holdings, Inc. We appreciate Dr. Kazuya Omi, Dr. Hiroshi Hayashi and Dr. John Clyde Co Soriano for their critical review of this paper. We are also grateful to all volunteer blood donors. We also appreciate Mr. Masatoshi Mori, Mr. Takanori Endo, Mr. Daichi Takizawa, Ms. Yuko Sekiya, Mr. Yuta Takeda, Dr. Ryo Negishi, and Dr. Kyohei Yoshimitsu for their technical advice and assistance.

## Author contributions

**Conceptualization:** Tomoyuki Yoshida, Takeyuki Kotaka.

**Data curation:** Tomoyuki Yoshida, Yoshiaki Sakamoto, Aya Tsuruta, Nozomi Shiozawa.

**Formal analysis:** Tomoyuki Yoshida, Yoshiaki Sakamoto, Risa Kimura, Nozomi Shiozawa.

**Investigation:** Tomoyuki Yoshida.

**Methodology:** Tomoyuki Yoshida, Yoshiaki Sakamoto, Risa Kimura, Nozomi Shiozawa.

**Project administration:** Takeyuki Kotaka.

**Software:** Aya Tsuruta.

**Supervision:** Takeyuki Kotaka.

**Visualization:** Aya Tsuruta.

**Writing – original draft:** Tomoyuki Yoshida.

**Writing – review & editing:** Tomoyuki Yoshida, Yoshiaki Sakamoto, Nozomi Shiozawa.

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
