## [Decision Letter · Decision Letter 0]

10 Jul 2025

Dear Dr. Yoshida,

Thank you for submitting your manuscript to PLOS ONE. After careful consideration, we feel that it has merit but does not fully meet PLOS ONE’s publication criteria as it currently stands. Therefore, we invite you to submit a revised version of the manuscript that addresses the points raised during the review process.

Please address before the issues raised by reviewers...

We look forward to receiving your revised manuscript.

Kind regards,

Jeffrey Chalmers, Ph.D.

Academic Editor

PLOS ONE

Journal Requirements:

2. We note that your Data Availability Statement is currently as follows: [If the data are all contained within the manuscript and/or Supporting Information files, enter the following: All relevant data are within the manuscript and its Supporting Information files.]

[The authors have declared that no competing interests exist.]

We note that one or more of the authors are employed by a commercial company: H.U. Group Research Institute, H.U. Cells, Inc.

              1. Please provide an amended Funding Statement declaring this commercial affiliation, as well as a statement regarding the Role of Funders in your study. If the funding organization did not play a role in the study design, data collection and analysis, decision to publish, or preparation of the manuscript and only provided financial support in the form of authors' salaries and/or research materials, please review your statements relating to the author contributions, and ensure you have specifically and accurately

indicated the role(s) that these authors had in your study. You can update author roles in the Author Contributions section of the online submission form.

Within your Competing Interests Statement, please confirm that this commercial affiliation does not alter your adherence to all PLOS ONE policies on sharing data and materials by including the following statement: "This does not alter our adherence to  PLOS ONE policies on sharing data and materials.” (as detailed online in our guide for authors http://journals.plos.org/plosone/s/competing-interests) . If this adherence statement is not accurate and  there are restrictions on sharing of data and/or materials, please state these. Please note that we cannot proceed with consideration of your article until this information has been declared

Additional Editor Comments:

Reviewers' comments:

Reviewer's Responses to Questions

**Comments to the Author**

1. Is the manuscript technically sound, and do the data support the conclusions?

Reviewer #1: Partly

Reviewer #2: Partly

Reviewer #3: Partly

2. Has the statistical analysis been performed appropriately and rigorously?

Reviewer #1: No

Reviewer #2: I Don't Know

Reviewer #3: No

3. Have the authors made all data underlying the findings in their manuscript fully available?

Reviewer #1: Yes

Reviewer #2: Yes

Reviewer #3: Yes

4. Is the manuscript presented in an intelligible fashion and written in standard English?

Reviewer #1: Yes

Reviewer #2: Yes

Reviewer #3: Yes

Reviewer #1: This manuscript addresses an important and timely topic—developing a novel centrifugation-based technique to isolate viable PBMCs from whole blood with minimal WBC and RBC contamination. The authors provide a clear description of current commercial techniques, particularly density gradient centrifugation, and highlight the limitations associated with these existing methods. The manuscript compares the performance of the proposed method against two widely used commercial technologies and includes preliminary data suggesting improved performance. However, several major revisions are necessary before the manuscript can be considered for publication.

1. The authors state that their method performs better than the current technologies, but the abstract and conclusion sections do not include any quantitative data to support this claim. Specific numerical results should be presented to demonstrate the comparative performance clearly.

2. The conclusions are somewhat overstated given the scope and depth of the current data. The authors should discuss in more detail the limitations of their study and future research that should be conducted.

3. While the manuscript provides some quantitative data, there is no statistical analysis presented to validate the claim that their technology is superior. Mean values and standard deviations should be reported for all comparative data, and appropriate statistical tests (e.g., ANOVA, t-tests) should be used to evaluate the significance of differences between groups.

4. Some suggestions for modifying the figures:

Figure 2: A schematic diagram comparing the protocols of FlowMagic, Lymphoprep, and SepMate would enhance clarity and help readers understand the procedural differences.

Figure 3: The upper, middle, and lower layers should be clearly labeled.

Figure 4 and 5: Data should include mean ± standard deviation, and statistical significance between groups (if any) should be indicated.

Reviewer #2: The article describes the development and evaluation of FlowMagic™, a new two-layer insert device for isolating human peripheral blood mononuclear cells (PBMCs) from whole blood, aiming to reduce red blood cell (RBC) and granulocyte (GRA) contamination commonly encountered in conventional methods. The device was compared with existing techniques (SepMate and Lymphoprep). The study addresses a significant technical challenge in the field and could be a useful tool for both research and medical laboratories, especially given the ongoing issue of RBC and GRA contamination when delayed processing is necessary due to sample transport or logistical factors.

Major Points for Improvement and Clarification:

• The experiments seem to include five volunteer donors, which is a reasonable starting point for a proof-of-concept study. However, the manuscript should clearly state whether any statistical analyses were performed, and if so, specify which tests were used, exact p-values, and confidence intervals, particularly given the sample size. This would strengthen the validity of the findings.

Specific comment:

o Was any statistical test performed for the comparisons described in the results (e.g., lines 140–141 and 148–153 regarding granulocyte contamination)? If so, please specify the tests and results in the text or figure legends.

• While the referenced figures (e.g., Fig. 4 and 5) demonstrate improvements, including actual numerical data, summary tables in the Results section would facilitate understanding and independent assessment. Mean values ± standard deviation, exact cell counts, and percentage improvements or fold-reduction in contamination would make the benefits more explicit.

Specific comments:

o In lines 136–138, it is stated that RBC contamination for all three methods increased after 24 and 72 hr, but Fig. 4d–f does not show an increase for FlowMagic™. How was the increase in contamination for FlowMagic™ determined? Please clarify.

o From lines 138–140, the manuscript mentions that the FlowMagic™ method excludes RBC contamination. Please clarify how this was determined, especially in light of the previous statement about increased contamination.

• More information on the time required and ease of use of the FlowMagic™ device would be helpful. A brief comparison with the workflow of existing devices would also be valuable for readers considering adopting this technology.

Minor Comments:

• In line 94, the question mark should be removed, and the parenthesis with “brand” should be replaced with the actual brand name.

Reviewer #3: Dear Authors,

Thank you for submitting your manuscript entitled “Development of a New Cell Isolation Device FlowMagic™” for consideration. Your work presents a timely and relevant advancement in the field of peripheral blood mononuclear cell (PBMC) isolation, particularly addressing the well-known challenges of red blood cell (RBC) and granulocyte (GRA) contamination in delayed blood processing. I commend your efforts in developing the FlowMagic™ device, and your comprehensive side-by-side comparison with SepMate and standard Lymphoprep techniques is a valuable contribution to the field.

Below, I offer a detailed critique aimed at enhancing the clarity, rigour, and impact of your manuscript.

Major Comments

1. Statistical Analysis and Rigor

While your data are promising, all findings are currently presented without formal statistical analysis. This is a significant limitation. For a manuscript to meet the standards of PLOS ONE, particularly when making comparative claims, appropriate statistical testing (e.g., ANOVA with post-hoc tests, paired t-tests, or non-parametric equivalents) must be applied.

Action: Please include appropriate statistical analyses across timepoints and isolation methods for all quantitative data (e.g., cell recovery, contamination levels, flow cytometry results). Report p-values and define significance thresholds in the methods section.

2. Sample Size and Justification

The study includes five donors, but no rationale is given for the chosen sample size, and inter-donor variability is not discussed in detail.

Please provide justification for the sample size, or consider increasing the number of donors. Additionally, clarify whether the same individuals were used across the three timepoints to control for inter-subject variability.

3. Figures 4 and 5 – Resolution and Readability

Thank you for including Figures 4 and 5, which provide critical support for your conclusions. However, both figures currently suffer from low resolution, limiting their interpretability:

Axis labels and tick marks are small and difficult to read.

No error bars or indicators of statistical variance are shown.

Legends do not consistently include units or definitions of abbreviations.

Data points are plotted but lack context (e.g., no median or confidence intervals).

Action: Please resubmit Figures 4 and 5 in high resolution (≥300 DPI). Ensure all panels include:

Clearly labeled axes (with units),

Error bars where applicable,

Statistical significance annotations (e.g., p < 0.05),

Expanded legends to enhance clarity.

Additionally, consider using consolidated bar graphs or boxplots instead of multiple dot plots, which may improve readability and facilitate comparisons.

4. Device Limitations and Overstatements

Your discussion makes strong claims regarding the superiority and novelty of FlowMagic™. While your data are promising, such claims should be more cautiously framed unless supported by broader benchmarking or independent validation.

Please temper phrases such as “this has not been resolved until now” and consider including limitations more explicitly. For example:

The inability of FlowMagic™ to isolate plasma simultaneously may be relevant for translational settings.

Functional downstream assays (e.g., cytokine release, ELISPOT) are not yet presented and should be acknowledged as needed future work.

5. Language and Editing

The manuscript would benefit from language polishing to improve clarity, precision, and readability. There are occasional grammatical issues and ambiguous terms (e.g., “corning tubes?” on pg. 8) that should be corrected.

I recommend engaging a professional scientific editor or fluent English speaker to review the entire text.

Minor Comments and Recommendations

Methods: Clarify if the same donor samples were used for all three timepoints across isolation methods.

Figures 1–2: The schematic CAD illustrations are useful; however, inclusion of dimensions would enhance technical comprehension.

References: The citations are appropriate and up to date. However, you may wish to add references to microfluidic PBMC isolation methods, which are emerging in this space and would provide valuable context for FlowMagic™'s niche.

Ethics: Approvals are clearly presented in both the manuscript and translated documents. Thank you for providing these in full.

Conclusion and Recommendation

Your study introduces a potentially impactful innovation in PBMC isolation, especially for delayed blood processing in clinical or field settings. The technical approach and device design are commendable, and your experimental comparisons are thoughtfully conceived.

That said, the manuscript currently requires major revisions to meet the scientific and editorial standards of PLOS ONE. In particular, the addition of statistical analysis, high-quality figures, and a more balanced discussion of strengths and limitations will substantially improve the manuscript’s impact and readability.

Once revised accordingly, I believe the work would make a valuable contribution to the literature on cell isolation technologies.

Thank you again for your submission and your thoughtful work in this important area.

**Do you want your identity to be public for this peer review?** For information about this choice, including consent withdrawal, please see our Privacy Policy

Reviewer #1: No

Reviewer #2: No

Reviewer #3: **Yes: ** Dr Panicos Shangaris

---

## [Author Response · Author response to Decision Letter 1]

8 Sep 2025

Response to reviewer 1

Reviewer #1: This manuscript addresses an important and timely topic—developing a novel centrifugation-based technique to isolate viable PBMCs from whole blood with minimal WBC and RBC contamination. The authors provide a clear description of current commercial techniques, particularly density gradient centrifugation, and highlight the limitations associated with these existing methods. The manuscript compares the performance of the proposed method against two widely used commercial technologies and includes preliminary data suggesting improved performance. However, several major revisions are necessary before the manuscript can be considered for publication.

1. The authors state that their method performs better than the current technologies, but the abstract and conclusion sections do not include any quantitative data to support this claim. Specific numerical results should be presented to demonstrate the comparative performance clearly.

Reply: We thank the reviewer for their comment. We have included specific numerical data, presenting median values with corresponding interquartile ranges, to provide clear evidence of comparative performance (Fig. 4-6, Table S1-S3). The results demonstrate that FlowMagic™ achieved significantly better outcomes than both SepMate and Lymphoprep methods, with statistical significance established at p < 0.01. (Line: 23-38)

2. The conclusions are somewhat overstated given the scope and depth of the current data. The authors should discuss in more detail the limitations of their study and future research that should be conducted.

Reply: We thank the reviewer for this valuable suggestion. We have addressed the above issue in the abstract section of our manuscript. We have revised it as follows: These findings indicate that the PBMC isolation method using FlowMagic™ is advantageous in the isolation process for research and laboratory tests to prevent contamination by RBCs and GRAs. (Lines: 39-41)

3. While the manuscript provides some quantitative data, there is no statistical analysis presented to validate the claim that their technology is superior. Mean values and standard deviations should be reported for all comparative data, and appropriate statistical tests (e.g., ANOVA, t-tests) should be used to evaluate the significance of differences between groups.

Reply: We increased the sample size to 10. For the data from a total of ten volunteers presented in Figures 4, 5, and 6, the Wilcoxon signed-rank test was used with Bonferroni correction for multiple comparisons as a non-parametric alternative. All data were visualized using box plots. A significance level of 0.05 was used for all statistical tests. In response to your recommendation, we have included detailed summary tables in the Supporting Information (Tables S1-S3) that present mean values ± standard deviation. (Figs 4, 5, and 6, table S1-S3)

4. Some suggestions for modifying the figures:

Figure 2: A schematic diagram comparing the protocols of FlowMagic, Lymphoprep, and SepMate would enhance clarity and help readers understand the procedural differences.

Reply: We thank the reviewer for this valuable suggestion. We have addressed this concern by incorporating a comprehensive schematic comparison of the three protocols in Figure 2b. This diagram clearly illustrates the procedural differences between FlowMagic�, Lymphoprep, and SepMate protocols, including the key steps of density gradient centrifugation, and cell collection phases. The visual representation in Figure 2b provides readers with an immediate understanding of the methodological distinctions between these three approaches, thereby enhancing the clarity and accessibility of the comparative analysis as requested by the reviewer (Fig 2).

Figure 3: The upper, middle, and lower layers should be clearly labeled.

Reply: We thank the reviewer for this important suggestion regarding layer identification. In response to this comment, we have added clear labels for the upper, middle, and lower layers in Figure 3a, 3b, and 3c. These labels now provide precise identification of each density gradient layer, facilitating better understanding of the spatial distribution of different cell populations and debris throughout the separation process. This enhancement improves the interpretability of the results and allows readers to more effectively correlate the visual observations with the quantitative data presented in the manuscript. (Fig 3)

Figure 4 and 5: Data should include mean ± standard deviation, and statistical significance between groups (if any) should be indicated.

Reply: For the data from a total of ten volunteers presented in Figures 4, 5, and 6, the Wilcoxon signed-rank test was used with Bonferroni correction for multiple comparisons as a non-parametric alternative. All data were visualized using box plots. A significance level of 0.05 was used for all statistical tests. In response to your recommendation, we have included detailed summary tables in the Supporting Information (Tables S1-S3) that present mean values ± standard deviation and p-value. (Figs 4, 5, and 6, table S1-S3).

Response to reviewer 2

Reviewer #2: The article describes the development and evaluation of FlowMagic™, a new two-layer insert device for isolating human peripheral blood mononuclear cells (PBMCs) from whole blood, aiming to reduce red blood cell (RBC) and granulocyte (GRA) contamination commonly encountered in conventional methods. The device was compared with existing techniques (SepMate and Lymphoprep). The study addresses a significant technical challenge in the field and could be a useful tool for both research and medical laboratories, especially given the ongoing issue of RBC and GRA contamination when delayed processing is necessary due to sample transport or logistical factors.

Major Points for Improvement and Clarification:

• The experiments seem to include five volunteer donors, which is a reasonable starting point for a proof-of-concept study. However, the manuscript should clearly state whether any statistical analyses were performed, and if so, specify which tests were used, exact p-values, and confidence intervals, particularly given the sample size. This would strengthen the validity of the findings.

Specific comment:

• Was any statistical test performed for the comparisons described in the results (e.g., lines 140–141 and 148–153 regarding granulocyte contamination)? If so, please specify the tests and results in the text or figure legends.

Reply: An additional validation study was conducted using samples from five volunteers to increase the sample size to a total of 10. For the data from ten volunteers presented in Figures 4, 5, and 6, Wilcoxon signed-rank test was used with Bonferroni correction for multiple comparisons as a non-parametric alternative. All data were visualized using box plots. A significance level of 0.05 was considered statistically significant for all statistical tests. In response to this comment, we have added the above information in the figure legends. (Lines: 182-231, Figs 4 and 5.)

• While the referenced figures (e.g., Fig. 4 and 5) demonstrate improvements, including actual numerical data, summary tables in the Results section would facilitate understanding and independent assessment. Mean values ± standard deviation, exact cell counts, and percentage improvements or fold-reduction in contamination would make the benefits more explicit.

Reply: Thank you for this valuable suggestion. We agree that providing comprehensive numerical data would greatly enhance the clarity and interpretability of our findings. In response to your recommendation, we have included detailed summary tables in the Supporting Information (Tables S1-S3) that present mean values ± standard deviation, exact cell counts, and percentage improvements for all three isolation methods. These tables provide the quantitative data necessary for independent assessment of the methodological improvements and contamination reduction achieved with each approach. The inclusion of these comprehensive datasets in the Supporting Information directly addresses your concern about facilitating understanding while maintaining the flow of the main Results section. We believe these additional data will allow readers to fully evaluate the magnitude of improvements demonstrated in our comparative analysis. (Lines: 182-260, Table S1-S3)

Specific comments:

• In lines 136–138, it is stated that RBC contamination for all three methods increased after 24 and 72 hr, but Fig. 4d–f does not show an increase for FlowMagic™. How was the increase in contamination for FlowMagic™ determined? Please clarify.

Reply: We thank the reviewer for this valuable suggestion. The aforementioned description has been modified as shown below. We recognize that the previous statement may have caused confusion. At 48 and 72 hours after blood collection, the results showed an increase in the amount of RBC contamination among the two methods except FlowMagic� (Figs 3b-c and 4b). (Lines: 182-204)

• From lines 138–140, the manuscript mentions that the FlowMagic™ method excludes RBC contamination. Please clarify how this was determined, especially in light of the previous statement about increased contamination.

Reply: We thank the reviewer for this valuable suggestion. The aforementioned description has been modified as shown below. We recognize that the previous statement may have caused confusion. Although FlowMagic� yielded less WBC than other methods for blood samples that were incubated for 48-72 hours, it was able to exclude RBC contamination and greatly reduce granulocyte contamination (Fig. 4b). (Lines: 182-204)

• More information on the time required and ease of use of the FlowMagic™ device would be helpful. A brief comparison with the workflow of existing devices would also be valuable for readers considering adopting this technology.

Reply: We thank the reviewer for this valuable suggestion. We have addressed this concern by incorporating a comprehensive schematic comparison of the three protocols in Figure 2b. This diagram clearly illustrates the procedural differences between FlowMagic�, Lymphoprep, and SepMate protocols, including the key steps of density gradient centrifugation, and cell collection phases. The visual representation in Figure 2b provides readers with an immediate understanding of the methodological distinctions between these three approaches, thereby enhancing the clarity and accessibility of the comparative analysis as requested by the reviewer (Fig. 2).

Minor Comments:

• In line 94, the question mark should be removed, and the parenthesis with “brand” should be replaced with the actual brand name.

Reply: Thank you very much for your comments. We have already addressed this issue. (Lines: 106)

Response to reviewer 3

Reviewer #3: Dear Authors,

Thank you for submitting your manuscript entitled “Development of a New Cell Isolation Device FlowMagic™” for consideration. Your work presents a timely and relevant advancement in the field of peripheral blood mononuclear cell (PBMC) isolation, particularly addressing the well-known challenges of red blood cell (RBC) and granulocyte (GRA) contamination in delayed blood processing. I commend your efforts in developing the FlowMagic™ device, and your comprehensive side-by-side comparison with SepMate and standard Lymphoprep techniques is a valuable contribution to the field.

Below, I offer a detailed critique aimed at enhancing the clarity, rigour, and impact of your manuscript.

Major Comments

1. Statistical Analysis and Rigor

While your data are promising, all findings are currently presented without formal statistical analysis. This is a significant limitation. For a manuscript to meet the standards of PLOS ONE, particularly when making comparative claims, appropriate statistical testing (e.g., ANOVA with post-hoc tests, paired t-tests, or non-parametric equivalents) must be applied.

Action: Please include appropriate statistical analyses across timepoints and isolation methods for all quantitative data (e.g., cell recovery, contamination levels, flow cytometry results). Report p-values and define significance thresholds in the methods section.

Reply: An additional validation study was conducted using samples from five volunteers. For the data from a total of ten volunteers presented in Figures 4, 5, and 6, the Wilcoxon signed-rank test was used with Bonferroni correction for multiple comparisons as a non-parametric alternative. All data were visualized using box plots. A value of 0.05 was considered significant for all statistical tests. In response to your recommendation, we have included detailed summary tables in the Supporting Information (Tables S1-S3) that present mean values ± standard deviation, exact cell counts, and percentage improvements for all three isolation methods. (Figs 4, 5, and 6, table S1-S3)

2. Sample Size and Justification

The study includes five donors, but no rationale is given for the chosen sample size, and inter-donor variability is not discussed in detail.

Please provide justification for the sample size, or consider increasing the number of donors. Additionally, clarify whether the same individuals were used across the three timepoints to control for inter-subject variability.

Reply: An additional validation study was conducted using samples from five volunteers to increase the sample size to ten. For the data from a total of ten volunteers presented in Figures 4, 5, and 6, Wilcoxon signed-rank test was used with Bonferroni correction for multiple comparisons as a non-parametric alternative. All data were visualized using box plots. A significance level of 0.05 was used for all statistical tests. Parallel PBMC isolations were performed by three different techniques within 24, 48, and 72 hours after blood collection from the same volunteers. (Line: 102-103)

3. Figures 4 and 5 – Resolution and Readability

Thank you for including Figures 4 and 5, which provide critical support for your conclusions. However, both figures currently suffer from low resolution, limiting their interpretability:

Axis labels and tick marks are small and difficult to read.

No error bars or indicators of statistical variance are shown.

Legends do not consistently include units or definitions of abbreviations.

Data points are plotted but lack context (e.g., no median or confidence intervals).

Action: Please resubmit Figures 4 and 5 in high resolution (≥300 DPI). Ensure all panels include:

Clearly labeled axes (with units),

Error bars where applicable,

Statistical significance annotations (e.g., p < 0.05),

Expanded legends to enhance clarity.

Additionally, consider using consolidated bar graphs or boxplots instead of multiple dot plots, which may improve readability and facilitate comparisons.

Reply: We thank the reviewer for this valuable suggestion. We have addressed this concern through improvements in Fig 4, 5, and 6. An additional validation study was conducted using samples from five volunteers. For the data from a total of ten volunteers presented in Figures 4, 5, and 6, the Wilcoxon signed-rank test was used with Bonferroni correction for multiple comparisons as a non-parametric alternative. All data were visualized using box plots. A significance level of 0.05 was used for all statistical tests.

4. Device Limitations and Overstatements

Your discussion makes strong claims regarding the superiority and novelty of FlowMagic™. While your data are promising, such claims should be more cautiously framed unless supported by broader benchmarking or independent validation.

Please temper phrases such as “this has not been resolved until now” and consider including limitations more explicitly. For example:

Reply: We thank the reviewer for this valuable suggestion. We have already deleted the above phrase from our manuscript.

The inability of FlowMagic™ to isolate plasma simultaneously may be relevant for translational settings.

Functional downstream assays (e.g., cytokine release, ELISPOT) are not yet presented and should be acknowledged as needed future work.

Reply: We thank the reviewer for this valuable suggestion. We have already added the above phrase into the discussion s

---

## [Decision Letter · Decision Letter 1]

6 Oct 2025

Development of a New Cell Isolation Device FlowMagicTM

PONE-D-25-20717R1

Dear Dr. Yoshida,

We’re pleased to inform you that your manuscript has been judged scientifically suitable for publication and will be formally accepted for publication once it meets all outstanding technical requirements.

Kind regards,

Jeffrey Chalmers, Ph.D.

Academic Editor

PLOS ONE

Additional Editor Comments (optional):

Reviewers' comments:

Reviewer's Responses to Questions

**Comments to the Author**

Reviewer #1: All comments have been addressed

Reviewer #2: All comments have been addressed

2. Is the manuscript technically sound, and do the data support the conclusions?

Reviewer #1: Yes

Reviewer #2: Yes

3. Has the statistical analysis been performed appropriately and rigorously?

Reviewer #1: Yes

Reviewer #2: Yes

4. Have the authors made all data underlying the findings in their manuscript fully available?

Reviewer #1: Yes

Reviewer #2: Yes

5. Is the manuscript presented in an intelligible fashion and written in standard English?

Reviewer #1: Yes

Reviewer #2: Yes

Reviewer #1: The authors have addressed the reviewers' comments well. The reviewers have edited the figures and included appropriate statistical analysis to backup their claims.

Reviewer #2: (No Response)

**Do you want your identity to be public for this peer review?** For information about this choice, including consent withdrawal, please see our Privacy Policy

Reviewer #1: No

Reviewer #2: No

---

## [Editor Report · Acceptance letter]

PONE-D-25-20717R1

PLOS ONE

Dear Dr. Yoshida,

I'm pleased to inform you that your manuscript has been deemed suitable for publication in PLOS ONE. Congratulations! Your manuscript is now being handed over to our production team.

Kind regards,

on behalf of

Dr. Jeffrey Chalmers

Academic Editor

PLOS ONE